# Inductive Representation Learning on Large Graphs

**William L. Hamilton**$^*$
wleif@stanford.edu

**Rex Ying**$^*$
rexying@stanford.edu

**Jure Leskovec**
jure@cs.stanford.edu

Department of Computer Science
Stanford University
Stanford, CA, 94305

## Abstract

Low-dimensional embeddings of nodes in large graphs have proved extremely useful in a variety of prediction tasks, from content recommendation to identifying protein functions. However, most existing approaches require that all nodes in the graph are present during training of the embeddings; these previous approaches are inherently *transductive* and do not naturally generalize to unseen nodes. Here we present GraphSAGE, a general *inductive* framework that leverages node feature information (e.g., text attributes) to efficiently generate node embeddings for previously unseen data. Instead of training individual embeddings for each node, we learn a function that generates embeddings by sampling and aggregating features from a node's local neighborhood. Our algorithm outperforms strong baselines on three inductive node-classification benchmarks: we classify the category of unseen nodes in evolving information graphs based on citation and Reddit post data, and we show that our algorithm generalizes to completely unseen graphs using a multi-graph dataset of protein-protein interactions.

## 1   Introduction

Low-dimensional vector embeddings of nodes in large graphs[1] have proved extremely useful as feature inputs for a wide variety of prediction and graph analysis tasks [5, 11, 28, 35, 36]. The basic idea behind node embedding approaches is to use dimensionality reduction techniques to distill the high-dimensional information about a node's neighborhood into a dense vector embedding. These node embeddings can then be fed to downstream machine learning systems and aid in tasks such as node classification, clustering, and link prediction [11, 28, 35].

However, previous works have focused on embedding nodes from a single fixed graph, and many real-world applications require embeddings to be quickly generated for unseen nodes, or entirely new (sub)graphs. This inductive capability is essential for high-throughput, production machine learning systems, which operate on evolving graphs and constantly encounter unseen nodes (e.g., posts on Reddit, users and videos on Youtube). An inductive approach to generating node embeddings also facilitates generalization across graphs with the same form of features: for example, one could train an embedding generator on protein-protein interaction graphs derived from a model organism, and then easily produce node embeddings for data collected on new organisms using the trained model.

The inductive node embedding problem is especially difficult, compared to the transductive setting, because generalizing to unseen nodes requires "aligning" newly observed subgraphs to the node embeddings that the algorithm has already optimized on. An inductive framework must learn to

---

$^*$The two first authors made equal contributions.

[1]While it is common to refer to these data structures as social or biological *networks*, we use the term *graph* to avoid ambiguity with neural network terminology.

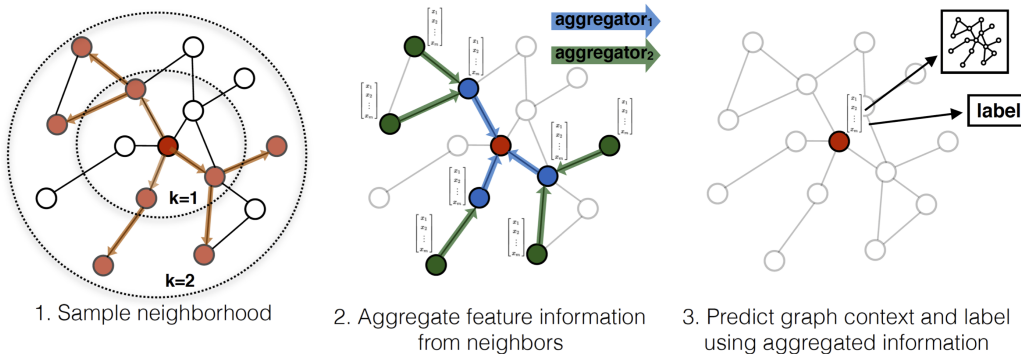

1. Sample neighborhood     2. Aggregate feature information     3. Predict graph context and label
from neighbors     using aggregated information

Figure 1: Visual illustration of the GraphSAGE sample and aggregate approach.

recognize structural properties of a node's neighborhood that reveal both the node's local role in the graph, as well as its global position.

Most existing approaches to generating node embeddings are inherently transductive. The majority of these approaches directly optimize the embeddings for each node using matrix-factorization-based objectives, and do not naturally generalize to unseen data, since they make predictions on nodes in a single, fixed graph [5, 11, 23, 28, 35, 36, 37, 39]. These approaches can be modified to operate in an inductive setting (e.g., [28]), but these modifications tend to be computationally expensive, requiring additional rounds of gradient descent before new predictions can be made. There are also recent approaches to learning over graph structures using convolution operators that offer promise as an embedding methodology [17]. So far, graph convolutional networks (GCNs) have only been applied in the transductive setting with fixed graphs [17, 18]. In this work we both extend GCNs to the task of inductive unsupervised learning and propose a framework that generalizes the GCN approach to use trainable aggregation functions (beyond simple convolutions).

**Present work**. We propose a general framework, called GraphSAGE (SAmple and aggreGatE), for inductive node embedding. Unlike embedding approaches that are based on matrix factorization, we leverage node features (e.g., text attributes, node profile information, node degrees) in order to learn an embedding function that generalizes to unseen nodes. By incorporating node features in the learning algorithm, we simultaneously learn the topological structure of each node's neighborhood as well as the distribution of node features in the neighborhood. While we focus on feature-rich graphs (e.g., citation data with text attributes, biological data with functional/molecular markers), our approach can also make use of structural features that are present in all graphs (e.g., node degrees). Thus, our algorithm can also be applied to graphs without node features.

Instead of training a distinct embedding vector for each node, we train a set of *aggregator functions* that learn to aggregate feature information from a node's local neighborhood (Figure 1). Each aggregator function aggregates information from a different number of hops, or search depth, away from a given node. At test, or inference time, we use our trained system to generate embeddings for entirely unseen nodes by applying the learned aggregation functions. Following previous work on generating node embeddings, we design an unsupervised loss function that allows GraphSAGE to be trained without task-specific supervision. We also show that GraphSAGE can be trained in a fully supervised manner.

We evaluate our algorithm on three node-classification benchmarks, which test GraphSAGE's ability to generate useful embeddings on unseen data. We use two evolving document graphs based on citation data and Reddit post data (predicting paper and post categories, respectively), and a multi-graph generalization experiment based on a dataset of protein-protein interactions (predicting protein functions). Using these benchmarks, we show that our approach is able to effectively generate representations for unseen nodes and outperform relevant baselines by a significant margin: across domains, our supervised approach improves classification F1-scores by an average of 51% compared to using node features alone and GraphSAGE consistently outperforms a strong, transductive baseline [28], despite this baseline taking $\sim 100\times$ longer to run on unseen nodes. We also show that the new aggregator architectures we propose provide significant gains (7.4% on average) compared to an aggregator inspired by graph convolutional networks [17]. Lastly, we probe the expressive capability of our approach and show, through theoretical analysis, that GraphSAGE is capable of learning structural information about a node's role in a graph, despite the fact that it is inherently based on features (Section 5).

## 2   Related work

Our algorithm is conceptually related to previous node embedding approaches, general supervised approaches to learning over graphs, and recent advancements in applying convolutional neural networks to graph-structured data.[2]

**Factorization-based embedding approaches**.   There are a number of recent node embedding approaches that learn low-dimensional embeddings using random walk statistics and matrix factorization-based learning objectives [5, 11, 28, 35, 36].  These methods also bear close relationships to more classic approaches to spectral clustering [23], multi-dimensional scaling [19], as well as the PageRank algorithm [25].  Since these embedding algorithms directly train node embeddings for individual nodes, they are inherently transductive and, at the very least, require expensive additional training (e.g., via stochastic gradient descent) to make predictions on new nodes. In addition, for many of these approaches (e.g., [11, 28, 35, 36]) the objective function is invariant to orthogonal transformations of the embeddings, which means that the embedding space does not naturally generalize between graphs and can drift during re-training. One notable exception to this trend is the Planetoid-I algorithm introduced by Yang et al. [40], which is an inductive, embedding-based approach to semi-supervised learning. However, Planetoid-I does not use any graph structural information during inference; instead, it uses the graph structure as a form of regularization during training. Unlike these previous approaches, we leverage feature information in order to train a model to produce embeddings for unseen nodes.

**Supervised learning over graphs**.  Beyond node embedding approaches, there is a rich literature on supervised learning over graph-structured data.  This includes a wide variety of kernel-based approaches, where feature vectors for graphs are derived from various graph kernels (see [32] and references therein).  There are also a number of recent neural network approaches to supervised learning over graph structures [7, 10, 21, 31]. Our approach is conceptually inspired by a number of these algorithms. However, whereas these previous approaches attempt to classify entire graphs (or subgraphs), the focus of this work is generating useful representations for individual nodes.

**Graph convolutional networks**.  In recent years, several convolutional neural network architectures for learning over graphs have been proposed (e.g., [4, 9, 8, 17, 24]). The majority of these methods do not scale to large graphs or are designed for whole-graph classification (or both) [4, 9, 8, 24]. However, our approach is closely related to the graph convolutional network (GCN), introduced by Kipf et al. [17, 18]. The original GCN algorithm [17] is designed for semi-supervised learning in a transductive setting, and the exact algorithm requires that the full graph Laplacian is known during training. A simple variant of our algorithm can be viewed as an extension of the GCN framework to the inductive setting, a point which we revisit in Section 3.3.

## 3   Proposed method: GraphSAGE

The key idea behind our approach is that we learn how to aggregate feature information from a node's local neighborhood (e.g., the degrees or text attributes of nearby nodes). We first describe the GraphSAGE embedding generation (i.e., forward propagation) algorithm, which generates embeddings for nodes assuming that the GraphSAGE model parameters are already learned (Section 3.1).  We then describe how the GraphSAGE model parameters can be learned using standard stochastic gradient descent and backpropagation techniques (Section 3.2).

### 3.1   Embedding generation (i.e., forward propagation) algorithm

In this section, we describe the embedding generation, or forward propagation algorithm (Algorithm 1), which assumes that the model has already been trained and that the parameters are fixed.  In particular, we assume that we have learned the parameters of $K$ aggregator functions (denoted $\textsc{aggregate}_k, \forall k \in \{1, ..., K\}$), which aggregate information from node neighbors, as well as a set of weight matrices $\mathbf{W}^k, \forall k \in \{1, ..., K\}$, which are used to propagate information between different layers of the model or "search depths". Section 3.2 describes how we train these parameters.

**Algorithm 1:** GraphSAGE embedding generation (i.e., forward propagation) algorithm

---

**Input** : Graph $\mathcal{G}(\mathcal{V}, \mathcal{E})$; input features $\{\mathbf{x}_v, \forall v \in \mathcal{V}\}$; depth $K$; weight matrices
$\mathbf{W}^k, \forall k \in \{1, ..., K\}$; non-linearity $\sigma$; differentiable aggregator functions
$\text{AGGREGATE}_k, \forall k \in \{1, ..., K\}$; neighborhood function $\mathcal{N} : v \to 2^{\mathcal{V}}$

**Output :** Vector representations $\mathbf{z}_v$ for all $v \in \mathcal{V}$

**1** $\mathbf{h}_v^0 \leftarrow \mathbf{x}_v, \forall v \in \mathcal{V}$ ;
**2 for** $k = 1...K$ **do**
**3**     **for** $v \in \mathcal{V}$ **do**
**4**        $\mathbf{h}_{\mathcal{N}(v)}^k \leftarrow \text{AGGREGATE}_k(\{\mathbf{h}_u^{k-1}, \forall u \in \mathcal{N}(v)\})$;
**5**        $\mathbf{h}_v^k \leftarrow \sigma\left(\mathbf{W}^k \cdot \text{CONCAT}(\mathbf{h}_v^{k-1}, \mathbf{h}_{\mathcal{N}(v)}^k)\right)$
**6**     **end**
**7**     $\mathbf{h}_v^k \leftarrow \mathbf{h}_v^k / \|\mathbf{h}_v^k\|_2, \forall v \in \mathcal{V}$
**8 end**
**9** $\mathbf{z}_v \leftarrow \mathbf{h}_v^K, \forall v \in \mathcal{V}$

---

The intuition behind Algorithm 1 is that at each iteration, or search depth, nodes aggregate information from their local neighbors, and as this process iterates, nodes incrementally gain more and more information from further reaches of the graph.

Algorithm 1 describes the embedding generation process in the case where the entire graph, $\mathcal{G} = (\mathcal{V}, \mathcal{E})$, and features for all nodes $\mathbf{x}_v, \forall v \in \mathcal{V}$, are provided as input. We describe how to generalize this to the minibatch setting below. Each step in the outer loop of Algorithm 1 proceeds as follows, where $k$ denotes the current step in the outer loop (or the depth of the search) and $\mathbf{h}^k$ denotes a node's representation at this step: First, each node $v \in \mathcal{V}$ aggregates the representations of the nodes in its immediate neighborhood, $\{\mathbf{h}_u^{k-1}, \forall u \in \mathcal{N}(v)\}$, into a single vector $\mathbf{h}_{\mathcal{N}(v)}^{k-1}$. Note that this aggregation step depends on the representations generated at the previous iteration of the outer loop (i.e., $k - 1$), and the $k = 0$ ("base case") representations are defined as the input node features. After aggregating the neighboring feature vectors, GraphSAGE then concatenates the node's current representation, $\mathbf{h}_v^{k-1}$, with the aggregated neighborhood vector, $\mathbf{h}_{\mathcal{N}(v)}^{k-1}$, and this concatenated vector is fed through a fully connected layer with nonlinear activation function $\sigma$, which transforms the representations to be used at the next step of the algorithm (i.e., $\mathbf{h}_v^k, \forall v \in \mathcal{V}$). For notational convenience, we denote the final representations output at depth $K$ as $\mathbf{z}_v \equiv \mathbf{h}_v^K, \forall v \in \mathcal{V}$. The aggregation of the neighbor representations can be done by a variety of aggregator architectures (denoted by the AGGREGATE placeholder in Algorithm 1), and we discuss different architecture choices in Section 3.3 below.

To extend Algorithm 1 to the minibatch setting, given a set of input nodes, we first forward sample the required neighborhood sets (up to depth $K$) and then we run the inner loop (line 3 in Algorithm 1), but instead of iterating over all nodes, we compute only the representations that are necessary to satisfy the recursion at each depth (Appendix A contains complete minibatch pseudocode).

**Relation to the Weisfeiler-Lehman Isomorphism Test**. The GraphSAGE algorithm is conceptually inspired by a classic algorithm for testing graph isomorphism. If, in Algorithm 1, we (i) set $K = |\mathcal{V}|$, (ii) set the weight matrices as the identity, and (iii) use an appropriate hash function as an aggregator (with no non-linearity), then Algorithm 1 is an instance of the Weisfeiler-Lehman (WL) isomorphism test, also known as "naive vertex refinement" [32]. If the set of representations $\{\mathbf{z}_v, \forall v \in \mathcal{V}\}$ output by Algorithm 1 for two subgraphs are identical then the WL test declares the two subgraphs to be isomorphic. This test is known to fail in some cases, but is valid for a broad class of graphs [32]. GraphSAGE is a continuous approximation to the WL test, where we replace the hash function with trainable neural network aggregators. Of course, we use GraphSAGE to generate useful node representations–not to test graph isomorphism. Nevertheless, the connection between GraphSAGE and the classic WL test provides theoretical context for our algorithm design to learn the topological structure of node neighborhoods.

**Neighborhood definition**. In this work, we uniformly sample a fixed-size set of neighbors, instead of using full neighborhood sets in Algorithm 1, in order to keep the computational footprint of each batch

fixed.[3] That is, using overloaded notation, we define $\mathcal{N}(v)$ as a fixed-size, uniform draw from the set $\{u \in \mathcal{V} : (u, v) \in \mathcal{E}\}$, and we draw different uniform samples at each iteration, $k$, in Algorithm 1. Without this sampling the memory and expected runtime of a single batch is unpredictable and in the worst case $O(|\mathcal{V}|)$. In contrast, the per-batch space and time complexity for GraphSAGE is fixed at $O(\prod_{i=1}^{K} S_i)$, where $S_i, i \in \{1, ..., K\}$ and $K$ are user-specified constants. Practically speaking we found that our approach could achieve high performance with $K = 2$ and $S_1 \cdot S_2 \leq 500$ (see Section 4.4 for details).

## 3.2 Learning the parameters of GraphSAGE

In order to learn useful, predictive representations in a fully unsupervised setting, we apply a graph-based loss function to the output representations, $\mathbf{z}_u, \forall u \in \mathcal{V}$, and tune the weight matrices, $\mathbf{W}^k, \forall k \in \{1, ..., K\}$, and parameters of the aggregator functions via stochastic gradient descent. The graph-based loss function encourages nearby nodes to have similar representations, while enforcing that the representations of disparate nodes are highly distinct:

$$J_{\mathcal{G}}(\mathbf{z}_u) = -\log\left(\sigma(\mathbf{z}_u^\top \mathbf{z}_v)\right) - Q \cdot \mathbb{E}_{v_n \sim P_n(v)} \log\left(\sigma(-\mathbf{z}_u^\top \mathbf{z}_{v_n})\right), \tag{1}$$

where $v$ is a node that co-occurs near $u$ on fixed-length random walk, $\sigma$ is the sigmoid function, $P_n$ is a negative sampling distribution, and $Q$ defines the number of negative samples. Importantly, unlike previous embedding approaches, the representations $\mathbf{z}_u$ that we feed into this loss function are generated from the features contained within a node's local neighborhood, rather than training a unique embedding for each node (via an embedding look-up).

This unsupervised setting emulates situations where node features are provided to downstream machine learning applications, as a service or in a static repository. In cases where representations are to be used only on a specific downstream task, the unsupervised loss (Equation 1) can simply be replaced, or augmented, by a task-specific objective (e.g., cross-entropy loss).

## 3.3 Aggregator Architectures

Unlike machine learning over N-D lattices (e.g., sentences, images, or 3-D volumes), a node's neighbors have no natural ordering; thus, the aggregator functions in Algorithm 1 must operate over an unordered set of vectors. Ideally, an aggregator function would be symmetric (i.e., invariant to permutations of its inputs) while still being trainable and maintaining high representational capacity. The symmetry property of the aggregation function ensures that our neural network model can be trained and applied to arbitrarily ordered node neighborhood feature sets. We examined three candidate aggregator functions:

**Mean aggregator**. Our first candidate aggregator function is the mean operator, where we simply take the elementwise mean of the vectors in $\{\mathbf{h}_u^{k-1}, \forall u \in \mathcal{N}(v)\}$. The mean aggregator is nearly equivalent to the convolutional propagation rule used in the transductive GCN framework [17]. In particular, we can derive an inductive variant of the GCN approach by replacing lines 4 and 5 in Algorithm 1 with the following:[4]

$$\mathbf{h}_v^k \leftarrow \sigma(\mathbf{W} \cdot \text{MEAN}(\{\mathbf{h}_v^{k-1}\} \cup \{\mathbf{h}_u^{k-1}, \forall u \in \mathcal{N}(v)\}). \tag{2}$$

We call this modified mean-based aggregator *convolutional* since it is a rough, linear approximation of a localized spectral convolution [17]. An important distinction between this convolutional aggregator and our other proposed aggregators is that it does not perform the concatenation operation in line 5 of Algorithm 1—i.e., the convolutional aggregator does concatenate the node's previous layer representation $\mathbf{h}_v^{k-1}$ with the aggregated neighborhood vector $\mathbf{h}_{\mathcal{N}(v)}^k$. This concatenation can be viewed as a simple form of a "skip connection" [13] between the different "search depths", or "layers" of the GraphSAGE algorithm, and it leads to significant gains in performance (Section 4).

**LSTM aggregator**. We also examined a more complex aggregator based on an LSTM architecture [14]. Compared to the mean aggregator, LSTMs have the advantage of larger expressive capability. However, it is important to note that LSTMs are not inherently symmetric (i.e., they are not permutation invariant), since they process their inputs in a sequential manner. We adapt LSTMs to operate on an unordered set by simply applying the LSTMs to a random permutation of the node's neighbors.

**Pooling aggregator**. The final aggregator we examine is both symmetric and trainable. In this *pooling* approach, each neighbor's vector is independently fed through a fully-connected neural network; following this transformation, an elementwise max-pooling operation is applied to aggregate information across the neighbor set:

$$\text{AGGREGATE}_k^{\text{pool}} = \max(\{\sigma\left(\mathbf{W}_{\text{pool}}\mathbf{h}_{u_i}^k + \mathbf{b}\right), \forall u_i \in \mathcal{N}(v)\}), \quad (3)$$

where $\max$ denotes the element-wise max operator and $\sigma$ is a nonlinear activation function. In principle, the function applied before the max pooling can be an arbitrarily deep multi-layer perceptron, but we focus on simple single-layer architectures in this work. This approach is inspired by recent advancements in applying neural network architectures to learn over general point sets [29]. Intuitively, the multi-layer perceptron can be thought of as a set of functions that compute features for each of the node representations in the neighbor set. By applying the max-pooling operator to each of the computed features, the model effectively captures different aspects of the neighborhood set. Note also that, in principle, any symmetric vector function could be used in place of the $\max$ operator (e.g., an element-wise mean). We found no significant difference between max- and mean-pooling in developments test and thus focused on max-pooling for the rest of our experiments.

## 4 Experiments

We test the performance of GraphSAGE on three benchmark tasks: (i) classifying academic papers into different subjects using the Web of Science citation dataset, (ii) classifying Reddit posts as belonging to different communities, and (iii) classifying protein functions across various biological protein-protein interaction (PPI) graphs. Sections 4.1 and 4.2 summarize the datasets, and the supplementary material contains additional information. In all these experiments, we perform predictions on nodes that are not seen during training, and, in the case of the PPI dataset, we test on entirely unseen graphs.

**Experimental set-up**. To contextualize the empirical results on our inductive benchmarks, we compare against four baselines: a random classifer, a logistic regression feature-based classifier (that ignores graph structure), the DeepWalk algorithm [28] as a representative factorization-based approach, and a concatenation of the raw features and DeepWalk embeddings. We also compare four variants of GraphSAGE that use the different aggregator functions (Section 3.3). Since, the "convolutional" variant of GraphSAGE is an extended, inductive version of Kipf et al's semi-supervised GCN [17], we term this variant GraphSAGE-GCN. We test unsupervised variants of GraphSAGE trained according to the loss in Equation (1), as well as supervised variants that are trained directly on classification cross-entropy loss. For all the GraphSAGE variants we used rectified linear units as the non-linearity and set $K = 2$ with neighborhood sample sizes $S_1 = 25$ and $S_2 = 10$ (see Section 4.4 for sensitivity analyses).

For the Reddit and citation datasets, we use "online" training for DeepWalk as described in Perozzi et al. [28], where we run a new round of SGD optimization to embed the new test nodes before making predictions (see the Appendix for details). In the multi-graph setting, we cannot apply DeepWalk, since the embedding spaces generated by running the DeepWalk algorithm on different disjoint graphs can be arbitrarily rotated with respect to each other (Appendix D).

All models were implemented in TensorFlow [1] with the Adam optimizer [16] (except DeepWalk, which performed better with the vanilla gradient descent optimizer). We designed our experiments with the goals of (i) verifying the improvement of GraphSAGE over the baseline approaches (i.e., raw features and DeepWalk) and (ii) providing a rigorous comparison of the different GraphSAGE aggregator architectures. In order to provide a fair comparison, all models share an identical implementation of their minibatch iterators, loss function and neighborhood sampler (when applicable). Moreover, in order to guard against unintentional "hyperparameter hacking" in the comparisons between GraphSAGE aggregators, we sweep over the same set of hyperparameters for all GraphSAGE variants (choosing the best setting for each variant according to performance on a validation set). The set of possible hyperparameter values was determined on early validation tests using subsets of the citation and Reddit data that we then discarded from our analyses. The appendix contains further implementation details.[5]

Table 1: Prediction results for the three datasets (micro-averaged F1 scores). Results for unsupervised and fully supervised GraphSAGE are shown. Analogous trends hold for macro-averaged scores.

| Name | Citation | | Reddit | | PPI | |
|---|---|---|---|---|---|---|
| | Unsup. F1 | Sup. F1 | Unsup. F1 | Sup. F1 | Unsup. F1 | Sup. F1 |
| Random | 0.206 | 0.206 | 0.043 | 0.042 | 0.396 | 0.396 |
| Raw features | 0.575 | 0.575 | 0.585 | 0.585 | 0.422 | 0.422 |
| DeepWalk | 0.565 | 0.565 | 0.324 | 0.324 | — | — |
| DeepWalk + features | 0.701 | 0.701 | 0.691 | 0.691 | — | — |
| GraphSAGE-GCN | 0.742 | 0.772 | **0.908** | 0.930 | 0.465 | 0.500 |
| GraphSAGE-mean | 0.778 | 0.820 | 0.897 | 0.950 | 0.486 | 0.598 |
| GraphSAGE-LSTM | 0.788 | 0.832 | **0.907** | **0.954** | 0.482 | **0.612** |
| GraphSAGE-pool | **0.798** | **0.839** | 0.892 | 0.948 | **0.502** | 0.600 |
| % gain over feat. | 39% | 46% | 55% | 63% | 19% | 45% |

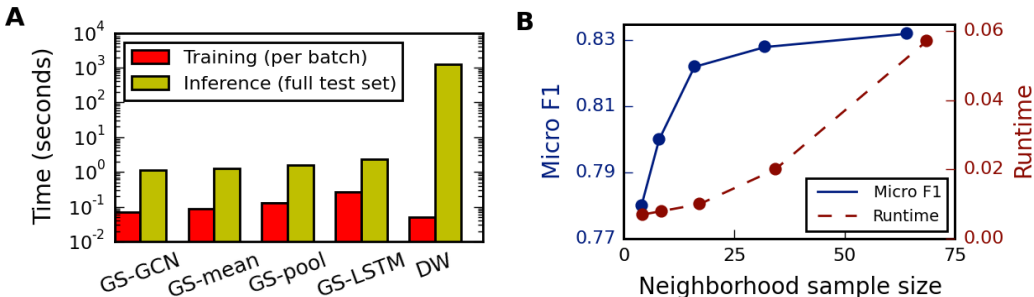

Figure 2: **A**: Timing experiments on Reddit data, with training batches of size 512 and inference on the full test set (79,534 nodes). **B**: Model performance with respect to the size of the sampled neighborhood, where the "neighborhood sample size" refers to the number of neighbors sampled at each depth for $K = 2$ with $S_1 = S_2$ (on the citation data using GraphSAGE-mean).

## 4.1 Inductive learning on evolving graphs: Citation and Reddit data

Our first two experiments are on classifying nodes in evolving information graphs, a task that is especially relevant to high-throughput production systems, which constantly encounter unseen data.

**Citation data**. Our first task is predicting paper subject categories on a large citation dataset. We use an undirected citation graph dataset derived from the Thomson Reuters Web of Science Core Collection, corresponding to all papers in six biology-related fields for the years 2000-2005. The node labels for this dataset correspond to the six different field labels. In total, this is dataset contains 302,424 nodes with an average degree of 9.15. We train all the algorithms on the 2000-2004 data and use the 2005 data for testing (with 30% used for validation). For features, we used node degrees and processed the paper abstracts according Arora et al.'s [2] sentence embedding approach, with 300-dimensional word vectors trained using the GenSim word2vec implementation [30].

**Reddit data**. In our second task, we predict which community different Reddit posts belong to. Reddit is a large online discussion forum where users post and comment on content in different topical communities. We constructed a graph dataset from Reddit posts made in the month of September, 2014. The node label in this case is the community, or "subreddit", that a post belongs to. We sampled 50 large communities and built a post-to-post graph, connecting posts if the same user comments on both. In total this dataset contains 232,965 posts with an average degree of 492. We use the first 20 days for training and the remaining days for testing (with 30% used for validation). For features, we use off-the-shelf 300-dimensional GloVe CommonCrawl word vectors [27]; for each post, we concatenated (i) the average embedding of the post title, (ii) the average embedding of all the post's comments (iii) the post's score, and (iv) the number of comments made on the post.

The first four columns of Table 1 summarize the performance of GraphSAGE as well as the baseline approaches on these two datasets. We find that GraphSAGE outperforms all the baselines by a significant margin, and the trainable, neural network aggregators provide significant gains compared

to the GCN approach. For example, the unsupervised variant GraphSAGE-pool outperforms the concatenation of the DeepWalk embeddings and the raw features by 13.8% on the citation data and 29.1% on the Reddit data, while the supervised version provides a gain of 19.7% and 37.2%, respectively. Interestingly, the LSTM based aggregator shows strong performance, despite the fact that it is designed for sequential data and not unordered sets. Lastly, we see that the performance of unsupervised GraphSAGE is reasonably competitive with the fully supervised version, indicating that our framework can achieve strong performance without task-specific fine-tuning.

## 4.2 Generalizing across graphs: Protein-protein interactions

We now consider the task of generalizing across graphs, which requires learning about node roles rather than community structure. We classify protein roles—in terms of their cellular functions from gene ontology—in various protein-protein interaction (PPI) graphs, with each graph corresponding to a different human tissue [41]. We use positional gene sets, motif gene sets and immunological signatures as features and gene ontology sets as labels (121 in total), collected from the Molecular Signatures Database [34]. The average graph contains 2373 nodes, with an average degree of 28.8. We train all algorithms on 20 graphs and then average prediction F1 scores on two test graphs (with two other graphs used for validation).

The final two columns of Table 1 summarize the accuracies of the various approaches on this data. Again we see that GraphSAGE significantly outperforms the baseline approaches, with the LSTM- and pooling-based aggregators providing substantial gains over the mean- and GCN-based aggregators.[6]

## 4.3 Runtime and parameter sensitivity

Figure 2.A summarizes the training and test runtimes for the different approaches. The training time for the methods are comparable (with GraphSAGE-LSTM being the slowest). However, the need to sample new random walks and run new rounds of SGD to embed unseen nodes makes DeepWalk $100\text{-}500\times$ slower at test time.

For the GraphSAGE variants, we found that setting $K = 2$ provided a consistent boost in accuracy of around 10-15%, on average, compared to $K = 1$; however, increasing $K$ beyond 2 gave marginal returns in performance (0-5%) while increasing the runtime by a prohibitively large factor of $10\text{-}100\times$, depending on the neighborhood sample size. We also found diminishing returns for sampling large neighborhoods (Figure 2.B). Thus, despite the higher variance induced by sub-sampling neighborhoods, GraphSAGE is still able to maintain strong predictive accuracy, while significantly improving the runtime.

## 4.4 Summary comparison between the different aggregator architectures

Overall, we found that the LSTM- and pool-based aggregators performed the best, in terms of both average performance and number of experimental settings where they were the top-performing method (Table 1). To give more quantitative insight into these trends, we consider each of the six different experimental settings (i.e., (3 datasets) × (unsupervised vs. supervised)) as trials and consider what performance trends are likely to generalize. In particular, we use the non-parametric Wilcoxon Signed-Rank Test [33] to quantify the differences between the different aggregators across trials, reporting the $T$-statistic and $p$-value where applicable. Note that this method is rank-based and essentially tests whether we would expect one particular approach to outperform another in a new experimental setting. Given our small sample size of only 6 different settings, this significance test is somewhat underpowered; nonetheless, the $T$-statistic and associated $p$-values are useful quantitative measures to assess the aggregators' relative performances.

We see that LSTM-, pool- and mean-based aggregators all provide statistically significant gains over the GCN-based approach ($T = 1.0$, $p = 0.02$ for all three). However, the gains of the LSTM and pool approaches over the mean-based aggregator are more marginal ($T = 1.5$, $p = 0.03$, comparing

LSTM to mean; $T = 4.5$, $p = 0.10$, comparing pool to mean). There is no significant difference between the LSTM and pool approaches ($T = 10.0$, $p = 0.46$). However, GraphSAGE-LSTM is significantly slower than GraphSAGE-pool (by a factor of $\approx 2\times$), perhaps giving the pooling-based aggregator a slight edge overall.

## 5   Theoretical analysis

In this section, we probe the expressive capabilities of GraphSAGE in order to provide insight into how GraphSAGE can learn about graph structure, even though it is inherently based on features. As a case-study, we consider whether GraphSAGE can learn to predict the clustering coefficient of a node, i.e., the proportion of triangles that are closed within the node's 1-hop neighborhood [38]. The clustering coefficient is a popular measure of how clustered a node's local neighborhood is, and it serves as a building block for many more complicated structural motifs [3]. We can show that Algorithm 1 is capable of approximating clustering coefficients to an arbitrary degree of precision:

**Theorem 1.** *Let* $\mathbf{x}_v \in U, \forall v \in \mathcal{V}$ *denote the feature inputs for Algorithm 1 on graph* $\mathcal{G} = (\mathcal{V}, \mathcal{E})$, *where* $U$ *is any compact subset of* $\mathbb{R}^d$. *Suppose that there exists a fixed positive constant* $C \in \mathbb{R}^+$ *such that* $\|\mathbf{x}_v - \mathbf{x}_{v'}\|_2 > C$ *for all pairs of nodes. Then we have that* $\forall \epsilon > 0$ *there exists a parameter setting* $\Theta^*$ *for Algorithm 1 such that after* $K = 4$ *iterations*

$$|z_v - c_v| < \epsilon, \forall v \in \mathcal{V},$$

*where* $z_v \in \mathbb{R}$ *are final output values generated by Algorithm 1 and* $c_v$ *are node clustering coefficients.*

Theorem 1 states that for any graph there exists a parameter setting for Algorithm 1 such that it can approximate clustering coefficients in that graph to an arbitrary precision, if the features for every node are distinct (and if the model is sufficiently high-dimensional). The full proof of Theorem 1 is in the Appendix. Note that as a corollary of Theorem 1, GraphSAGE can learn about local graph structure, even when the node feature inputs are sampled from an absolutely continuous random distribution (see the Appendix for details). The basic idea behind the proof is that if each node has a unique feature representation, then we can learn to map nodes to indicator vectors and identify node neighborhoods. The proof of Theorem 1 relies on some properties of the pooling aggregator, which also provides insight into why GraphSAGE-pool outperforms the GCN and mean-based aggregators.

## 6   Conclusion

We introduced a novel approach that allows embeddings to be efficiently generated for unseen nodes. GraphSAGE consistently outperforms state-of-the-art baselines, effectively trades off performance and runtime by sampling node neighborhoods, and our theoretical analysis provides insight into how our approach can learn about local graph structures. A number of extensions and potential improvements are possible, such as extending GraphSAGE to incorporate directed or multi-modal graphs. A particularly interesting direction for future work is exploring non-uniform neighborhood sampling functions, and perhaps even learning these functions as part of the GraphSAGE optimization.

**Acknowledgments**

The authors thank Austin Benson, Aditya Grover, Bryan He, Dan Jurafsky, Alex Ratner, Marinka Zitnik, and Daniel Selsam for their helpful discussions and comments on early drafts. The authors would also like to thank Ben Johnson for his many useful questions and comments on our code. This research has been supported in part by NSF IIS-1149837, DARPA SIMPLEX, Stanford Data Science Initiative, Huawei, and Chan Zuckerberg Biohub. W.L.H. was also supported by the SAP Stanford Graduate Fellowship and an NSERC PGS-D grant. The views and conclusions expressed in this material are those of the authors and should not be interpreted as necessarily representing the official policies or endorsements, either expressed or implied, of the above funding agencies, corporations, or the U.S. and Canadian governments.

## Footnotes

[2]In the time between this papers original submission to NIPS 2017 and the submission of the final, accepted (i.e., "camera-ready") version, there have been a number of closely related (e.g., follow-up) works published on pre-print servers. For temporal clarity, we do not review or compare against these papers in detail.

[3]Exploring non-uniform samplers is an important direction for future work.

[4]Note that this differs from Kipf et al's exact equation by a minor normalization constant [17].

[5]Code and links to the datasets: `http://snap.stanford.edu/graphsage/`

[6]Note that in very recent follow-up work Chen and Zhu [6] achieve superior performance by optimizing the GraphSAGE hyperparameters specifically for the PPI task and implementing new training techniques (e.g., dropout, layer normalization, and a new sampling scheme). We refer the reader to their work for the current state-of-the-art numbers on the PPI dataset that are possible using a variant of the GraphSAGE approach.

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
