[Supplementary Material]

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

[7]Note that these values differ from our previous reported pre-print values because they are corrected to account for an extraneous normalization by the batch size. We thank Ben Johnson for pointing out this discrepancy.

[8]`https://github.com/tensorflow/models/blob/master/tutorials/embedding/word2vec.py`

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

# Appendices

## A  Minibatch pseudocode

In order to use stochastic gradient descent, we adapt our algorithm to allow forward and backward propagation for minibatches of nodes and edges. Here we focus on the minibatch forward propagation algorithm, analogous to Algorithm 1. In the forward propagation of GraphSAGE the minibatch $\mathcal{B}$ contains nodes that we want to generate representations for. Algorithm 2 gives the pseudocode for the minibatch approach.

---

**Algorithm 2:** GraphSAGE minibatch forward propagation algorithm

> **Input** : Graph $\mathcal{G}(\mathcal{V}, \mathcal{E})$;
>   input features $\{\mathbf{x}_v, \forall v \in \mathcal{B}\}$;
>   depth $K$; weight matrices $\mathbf{W}^k, \forall k \in \{1, ..., K\}$;
>   non-linearity $\sigma$;
>   differentiable aggregator functions $\text{AGGREGATE}_k, \forall k \in \{1, ..., K\}$;
>   neighborhood sampling functions, $\mathcal{N}_k : v \to 2^{\mathcal{V}}, \forall k \in \{1, ..., K\}$
>
> **Output :** Vector representations $\mathbf{z}_v$ for all $v \in \mathcal{B}$

1  $\mathcal{B}^K \leftarrow \mathcal{B}$;
2  **for** $k = K...1$ **do**
3  $\quad$ $B^{k-1} \leftarrow \mathcal{B}^k$ ;
4  $\quad$ **for** $u \in \mathcal{B}^k$ **do**
5  $\quad\quad$ $\mathcal{B}^{k-1} \leftarrow \mathcal{B}^{k-1} \cup \mathcal{N}_k(u)$;
6  $\quad$ **end**
7  **end**
8  $\mathbf{h}_u^0 \leftarrow \mathbf{x}_v, \forall v \in \mathcal{B}^0$ ;
9  **for** $k = 1...K$ **do**
10 $\quad$ **for** $u \in \mathcal{B}^k$ **do**
11 $\quad\quad$ $\mathbf{h}_{\mathcal{N}(u)}^k \leftarrow \text{AGGREGATE}_k(\{\mathbf{h}_{u'}^{k-1}, \forall u' \in \mathcal{N}_k(u)\})$;
12 $\quad\quad$ $\mathbf{h}_u^k \leftarrow \sigma\left(\mathbf{W}^k \cdot \text{CONCAT}(\mathbf{h}_u^{k-1}, \mathbf{h}_{\mathcal{N}(u)}^k)\right)$;
13 $\quad\quad$ $\mathbf{h}_u^k \leftarrow \mathbf{h}_u^k / \|\mathbf{h}_u^k\|_2$;
14 $\quad$ **end**
15 **end**
16 $\mathbf{z}_u \leftarrow \mathbf{h}_u^K, \forall u \in \mathcal{B}$

---

The main idea is to sample all the nodes needed for the computation first. Lines 2-7 of Algorithm 2 correspond to the sampling stage. Each set $\mathcal{B}^k$ contains the nodes that are needed to compute the representations of nodes $v \in \mathcal{B}^{k+1}$, i.e., the nodes in the $(k + 1)$-st iteration, or "layer", of Algorithm 1. Lines 9-15 correspond to the aggregation stage, which is almost identical to the batch inference algorithm. Note that in Lines 12 and 13, the representation at iteration $k$ of any node in set $\mathcal{B}^k$ can be computed, because its representation at iteration $k - 1$ and the representations of its sampled neighbors at iteration $k - 1$ have already been computed in the previous loop. The algorithm thus avoids computing the representations for nodes that are not in the current minibatch and not used during the current iteration of stochastic gradient descent. We use the notation $\mathcal{N}_k(u)$ to denote a deterministic function which specifies a random sample of a node's neighborhood (i.e., the randomness is assumed to be pre-computed in the mappings). We index this function by $k$ to denote the fact that the random samples are independent across iterations over $k$. We use a uniform sampling function in this work and sample with replacement in cases where the sample size is larger than the node's degree.

Note that the sampling process in Algorithm 2 is conceptually reversed compared to the iterations over $k$ in Algorithm 1: we start with the "layer-K" nodes (i.e., the nodes in $\mathcal{B}$) that we want to generate representations for; then we sample their neighbors (i.e., the nodes at "layer-K-1" of the algorithm) and so on. One consequence of this is that the definition of neighborhood sampling sizes can be somewhat counterintuitive. In particular, if we use $K = 2$ total iterations with sample sizes $S_1$

and $S_2$ then this means that we sample $S_1$ nodes during iteration $k = 1$ of Algorithm 1 and $S_2$ nodes during iteration $k = 2$, and—from the perspective of the "target" nodes in $\mathcal{B}$ that we want to generate representations for after iteration $k = 2$—this amounts to sampling $S_2$ of their immediate neighbors and $S_1 \cdot S_2$ of their 2-hop neighbors.

## B  Additional Dataset Details

In this section, we provide some additional, relevant dataset details. The full PPI and Reddit datasets are available at: `http://snap.stanford.edu/graphsage/`. The Web of Science dataset (WoS) is licensed by Thomson Reuters and can be made available to groups with valid WoS licenses.

**Reddit data**   To sample communities, we ranked communities by their total number of comments in 2014 and selected the communities with ranks [11,50] (inclusive). We omitted the largest communities because they are large, generic default communities that substantially skew the class distribution. We selected the largest connected component of the graph defined over the union of these communities. We performed early validation experiments and model development on data from October and November, 2014.

Details on the source of the Reddit data are at: `https://archive.org/details/` `FullRedditSubmissionCorpus2006ThruAugust2015` and `https://archive.` `org/details/2015_reddit_comments_corpus`.

**WoS data**   We selected the following subfields manually, based on them being of relatively equal size and all biology-related fields. We performed early validation and model development on the neuroscience subfield (code=RU, which is excluded from our final set). We did not run any experiments on any other subsets of the WoS data. We took the largest connected component of the graph defined over the union of these fields.

- Immunology (code: NI, number of documents: 77356)
- Ecology (code: GU, number of documents: 37935)
- Biophysics (code: DA, number of documents: 36688)
- Endocrinology and Metabolism (code: IA, number of documents: 52225).
- Cell Biology (code: DR, number of documents: 84231)
- Biology (other) (code: CU, number of documents: 13988)

**PPI Tissue Data**   For training, we randomly selected 20 PPI networks that had at least 15,000 edges. For testing and validation, we selected 4 large networks (2 for validation, 2 for testing, each with at least 35,000 edges). All experiments for model design and development were performed on the same 2 validation networks, and we used the same random training set in all experiments.

We selected features that included at least 10% of the proteins that appear in any of the PPI graphs. Note that the feature data is very sparse for dataset (42% of nodes have no non-zero feature values), which makes leveraging neighborhood information critical.

## C  Details on the Experimental Setup and Hyperparameter Tuning

**Random walks for the unsupervised objective**   For all settings, we ran 50 random walks of length 5 from each node in order to obtain the pairs needed for the unsupervised loss (Equation 1). Our implementation of the random walks is in pure Python and is based directly on Python code provided by Perozzi et al. [28].

**Logistic regression model**   For the feature only model and to make predictions on the embeddings output from the unsupervised models, we used the logistic SGDClassifier from the scikit-learn Python package [26], with all default settings. Note that this model is always optimized only on the training nodes and it is not fine-tuned on the embeddings that are generated for the test data.

**Hyperparameter selection** In all settings, we performed hyperparameter selection on the learning rate and the model dimension. With the exception of DeepWalk, we performed a parameter sweep on initial learning rates $\{0.01, 0.001, 0.0001\}$ for the supervised models and $\{2 \times 10^{-6}, 2 \times 10^{-7}, 2 \times 10^{-8}\}$ for the unsupervised models.[7] When applicable, we tested a "big" and "small" version of each model, where we tried to keep the overall model sizes comparable. For the pooling aggregator, the "big" model had a pooling dimension of 1024, while the "small" model had a dimension of 512. For the LSTM aggregator, the "big" model had a hidden dimension of 256, while the "small" model had a hidden dimension of 128; note that the actual parameter count for the LSTM is roughly $4\times$ this number, due to weights for the different gates. In all experiments and for all models we specify the output dimension of the $\mathbf{h}_i^k$ vectors at every depth $k$ of the recursion to be 256. All models use rectified linear units as a non-linear activation function. All the unsupervised GraphSAGE models and DeepWalk used 20 negative samples with context distribution smoothing over node degrees using a smoothing parameter of 0.75, following [11, 22, 28]. Initial experiments revealed that DeepWalk performed much better with large learning rates, so we swept over rates in the set $\{0.2, 0.4, 0.8\}$. For the supervised GraphSAGE methods, we ran 10 epochs for all models. All methods except DeepWalk use batch sizes of 512. We found that DeepWalk achieved faster wall-clock convergence with a smaller batch size of 64.

**Hardware** Except for DeepWalk, we ran experiments single a machine with 4 NVIDIA Titan X Pascal GPUs (12Gb of RAM at 10Gbps speed), 16 Intel Xeon CPUs (E5-2623 v4 @ 2.60GHz), and 256Gb of RAM. DeepWalk was faster on a CPU intensive machine with 144 Intel Xeon CPUs (E7-8890 v3 @ 2.50GHz) and 2Tb of RAM. Overall, our experiments took about 3 days in a shared resource setting. We expect that a consumer-grade single-GPU machine (e.g., with a Titan X GPU) could complete our full set of experiments in 4-7 days, if its full resources were dedicated.

**Notes on the DeepWalk implementation** Existing DeepWalk implementations [28, 11] are simply wrappers around dedicated word2vec code, and they do not easily support embedding new nodes and other variations. Moreover, this makes it difficult to compare runtimes and other statistics for these approaches. For this reason, we reimplemented DeepWalk in pure TensorFlow, using the vector initializations etc that are described in the TensorFlow word2vec tutorial.[8]

We found that DeepWalk was much slower to converge than the other methods, and since it is 2-5X faster at training, we gave it 5 passes over the random walk data, instead of one. To update the DeepWalk method on new data, we ran 50 random walks of length 5 (as described above) and performed updates on the embeddings for the new nodes while holding the already trained embeddings fixed. We also tested two variants, one where we restricted the sampled random walk "context nodes" to only be from the set of already trained nodes (which alleviates statistical drift) and an approach without this restriction. We always selected the better performing variant. Note that despite DeepWalk's poor performance on the inductive task, it is far more competitive when tested in the transductive setting, where it can be extensively trained on a single, fixed graph. (That said, Kipf et al [17][18] found that GCN-based approach consistently outperformed DeepWalk, even in the transductive setting on link prediction, a task that theoretically favors DeepWalk.) We did observe DeepWalk's performance *could* improve with further training, and in some cases it could become competitive with the unsupervised GraphSAGE approaches (but not the supervised approaches) if we let it run for $>1000\times$ longer than the other approaches (in terms of wall clock time for prediction on the test set); however, we did not deem this to be a meaningful comparison for the inductive task.

Note that DeepWalk is also equivalent to the node2vec model [11] with $p = q = 1$.

**Notes on neighborhood sampling** Due to the heavy-tailed nature of degree distributions we downsample the edges in all graphs before feeding them into the GraphSAGE algorithm. In particular, we subsample edges so that no node has degree larger than 128. Since we only sample at most 25 neighbors per node, this is a reasonable tradeoff. This downsampling allows us to store neighborhood information as dense adjacency lists, which drastically improves computational efficiency. For the Reddit data we also downsampled the edges of the original graph as a pre-processing step, since the

original graph is extremely dense. All experiments are on the downsampled version, but we release the full version on the project website for reference.

## D  Alignment Issues and Orthogonal Invariance for DeepWalk and Related Approaches

DeepWalk [28], node2vec [11], and other recent successful node embedding approaches employ objective functions of the form:

$$\alpha \sum_{i,j \in \mathcal{A}} f(\mathbf{z}_i^\top \mathbf{z}_j) + \beta \sum_{i,j \in \mathcal{B}} g(\mathbf{z}_i^\top \mathbf{z}_j) \tag{4}$$

where $f$, $g$ are smooth, continuous functions, $\mathbf{z}_i$ are the node representations that are being directly optimized (i.e., via embedding look-ups), and $\mathcal{A}, \mathcal{B}$ are sets of pairs of nodes. Note that in many cases, in the actual code implementations used by the authors of these approaches, nodes are associated with two unique embedding vectors and the arguments to the dot products in $f$ and $g$ are drawn for distinct embedding look-ups (e.g., [11, 28]); however, this does not fundamentally alter the learning algorithm. The majority of approaches also normalize the learned embeddings to unit length, so we assume this post-processing as well.

By connection to word embedding approaches and the arguments of [20], these approaches can also be viewed as stochastic, implicit matrix factorizations where we are trying to learn a matrix $\mathbf{Z} \in \mathbb{R}^{|\mathcal{V}| \times d}$ such that

$$\mathbf{Z}\mathbf{Z}^\top \approx \mathbf{M}, \tag{5}$$

where $\mathbf{M}$ is some matrix containing random walk statistics.

An important consequence of this structure is that the embeddings can be rotated by an arbitrary orthogonal matrix, without impacting the objective:

$$\mathbf{Z}\mathbf{Q}^\top \mathbf{Q}\mathbf{Z}^\top = \mathbf{Z}\mathbf{Z}^\top, \tag{6}$$

where $\mathbf{Q} \in \mathbb{R}^{d \times d}$ is any orthogonal matrix. Since the embeddings are otherwise unconstrained and the only error signal comes from the orthogonally-invariant objective (**??**), the entire embedding space is free to arbitrarily rotate during training.

Two clear consequences of this are:

1. Suppose we run an embedding approach based on (**??**) on two separate graphs A and B using the same output dimension. Without some explicit penalty enforcing alignment, the learned embeddings spaces for the two graphs will be arbitrarily rotated with respect to each other after training. Thus, for any node classification method that is trained on individual embeddings from graph A, inputting the embeddings from graph B will be essentially random. This fact is also simply true by virtue of the fact that the $\mathbf{M}$ matrices of these graphs are completely disjoint. Of course, if we had a way to match "similar" nodes between the graphs, then it could be possible to use an alignment procedure to share information between the graphs, such as the procedure proposed by [12] for aligning the output of word embedding algorithms. Investigating such alignment procedures is an interesting direction for future work; though these approaches will inevitably be slow run on new data, compared to approaches like GraphSAGE that can simply generate embeddings for new nodes without any additional training or alignment.

2. Suppose that we run an embedding approach based on (**??**) on graph C at time $t$ and train a classifier on the learned embeddings. Then at time $t + 1$ we add more nodes to C and run a new round of SGD and update all embeddings. Two issues arise: First by analogy to point 1 above, if the new nodes are only connected to a very small number of the old nodes, then the embedding space for the new nodes can essentially become rotated with respect to the original embedding space. Moreover, if we update all embeddings during training (not just for the new nodes), as suggested by [28]'s streaming approach to DeepWalk, then the embedding space can arbitrarily rotate compared to the embedding space that we trained our classifier on, which only further exasperates the problem.

Note that this rotational invariance is not problematic for tasks that only rely on pairwise node distances (e.g., link prediction via dot products). Moreover, some reasonable approaches to alleviate this issue of statistical drift are to (1) not update the already trained embeddings when optimizing the embeddings for new test nodes and (2) to only keep existing nodes as "context nodes" in the sampled random walks, i.e. to ensure that every dot-product in the skip-gram objective is the product of an already-trained node and a new/test node. We tried both of these approaches in this work and always selected the best performing DeepWalk variant.

Also note that empirically DeepWalk performs better on the citation data than the Reddit data (Section 4.1) because this statistical drift is worse in the Reddit data, compared to the citation graph. In particular, the Reddit data has fewer edges from the test set to the train set, which help prevent mis-alignment: 96% of the 2005 citation links connect back to the 2000-2004 data, while only 73% of edges in the Reddit test set connect back to the train data.

## E   Proof of Theorem 1

To prove Theorem 1, we first prove three lemmas:

- Lemma 1 states that there exists a continuous function that is guaranteed to only be positive in closed balls around a fixed number of points, with some noise tolerance.

- Lemma 2 notes that we can approximate the function in Lemma 1 to an arbitrary precision using a multilayer perceptron with a single hidden layer.

- Lemma 3 builds off the preceding two lemmas to prove that the pooling architecture can learn to map nodes to unique indicator vectors, assuming that all the input feature vectors are sufficiently distinct.

We also rely on fact that the max-pooling operator (with at least one hidden layer) is capable of approximating any Hausdorff continuous, symmetric function to an arbitrary $\epsilon$ precision [29].

We note that all of the following are essentially *identifiability* arguments. We show that there exists a parameter setting for which Algorithm 1 can learn nodes clustering coefficients, which is non-obvious given that it operates by aggregating feature information. The *efficient learnability* of the functions described is the subject of future work. We also note that these proofs are conservative in the sense that clustering coefficients may be in fact identifiable in fewer iterations, or with less restrictions, than we impose. Moreover, due to our reliance on two universal approximation theorems [15, 29], the required dimensionality is in principle $O(|\mathcal{V}|)$. We can provide a more informative bound on the required output dimension of some particular layers (e..g., Lemma 3); however, in the worst case this identifiability argument relies on having a dimension of $O(|\mathcal{V}|)$. It is worth noting, however, that Kipf et al's "featureless" GCN approach has parameter dimension $O(|\mathcal{V}|)$, so this requirement is not entirely unreasonable [17, 18].

Following Theorem 1, we let $\mathbf{x}_v \in U, \forall v \in \mathcal{V}$ denote the feature inputs for Algorithm 1 on graph $\mathcal{G} = (\mathcal{V}, \mathcal{E})$, where $U$ is any compact subset of $\mathbb{R}^d$.

**Lemma 1.** *Let $C \in \mathbb{R}^+$ be a fixed positive constant. Then for any non-empty finite subset of nodes $\mathcal{D} \subseteq \mathcal{V}$, there exists a continuous function $g : U \to \mathbb{R}$ such that*

$$\begin{cases} g(\mathbf{x}) > \epsilon, & \text{if } \|\mathbf{x} - \mathbf{x}_v\|_2 = 0 \text{ for some } v \in \mathcal{D} \\ g(\mathbf{x}) \leq -\epsilon, & \text{if } \|\mathbf{x} - \mathbf{x}_v\|_2 > C, \forall v \in \mathcal{D}, \end{cases} \tag{7}$$

*where $\epsilon < 0.5$ is a chosen error tolerance.*

*Proof.* Many such functions exist. For concreteness, we provide one construction that satisfies these criteria. Let $\mathbf{x} \in U$ denote an arbitrary input to $g$, let $d_v = \|\mathbf{x} - \mathbf{x}_v\|_2, \forall v \in \mathcal{D}$, and let $g$ be defined as $g(\mathbf{x}) = \sum_{v \in \mathcal{D}} g_v(\mathbf{x})$ with

$$g_v(\mathbf{x}) = \frac{3|\mathcal{D}|\epsilon}{bd_v^2 + 1} - 2\epsilon \tag{8}$$

where $b = \frac{3|\mathcal{D}|-1}{C^2} > 0$. By construction:

1. $g_v$ has a unique maximum of $3|\mathcal{D}|\epsilon - 2\epsilon > 2|\mathcal{D}|\epsilon$ at $d_v = 0$.

2. $\lim_{d_v \to \infty} \left( \frac{3|\mathcal{D}|\epsilon}{bd_v^2+1} - 2\epsilon \right) = -2\epsilon$

3. $\frac{3|\mathcal{D}|\epsilon}{bd_v^2+1} - 2\epsilon \leq -\epsilon$ if $d_v \geq C$.

Note also that $g$ is continuous on its domain ($d_v \in \mathbb{R}^+$) since it is the sum of finite set of continuous functions. Moreover, we have that, for a given input $\mathbf{x} \in U$, if $d_v \geq C$ for all points $v \in \mathcal{D}$ then $g(\mathbf{x}) = \sum_{v \in \mathcal{D}} g_v(\mathbf{a}) \leq -\epsilon$ by property 3 above. And, if $d_v = 0$ for any $v \in \mathcal{D}$, then $g$ is positive by construction, by properties 1 and 2, since in this case,

$$g_v(\mathbf{x}) + \sum_{v' \in \mathcal{D} \backslash v} g_{v'}(\mathbf{x}) \geq g_v(\mathbf{x}) - (|\mathcal{D}| - 1)2\epsilon$$
$$> g_v(\mathbf{x}) - 2(|\mathcal{D}|)\epsilon$$
$$> 2(|\mathcal{D}|)\epsilon - 2(|\mathcal{D}|)\epsilon$$
$$> 0,$$

so we know that $g$ is positive whenever $d_v = 0$ for any node and negative whenever $d_v > C$ for all nodes. □

**Lemma 2.** *The function $g : U \to \mathbb{R}$ can be approximated to an arbitrary degree of precision by standard multilayer perceptron (MLP) with least one hidden layer and a non-constant monotonically increasing activation function (e.g., a rectified linear unit). In precise terms, if we let $f_{\theta_\sigma}$ denote this MLP and $\theta_\sigma$ its parameters, we have that $\forall \epsilon, \exists \theta_\sigma$ such that $|f_{\theta_\sigma}(\mathbf{x}) - g(\mathbf{x})| < \epsilon|, \forall \mathbf{x} \in U$.*

*Proof.* This is a direct consequence of Theorem 2 in [15]. □

**Lemma 3.** *Let $\mathbf{A}$ be the adjacency matrix of $G$, let $\mathcal{N}^3(v)$ denote the 3-hop neighborhood of a node, $v$, and define $\chi(\mathcal{G}^3)$ as the chromatic number of the graph with adjacency matrix $\mathbf{A}^3$ (ignoring self-loops). Suppose that there exists a fixed positive constant $C \in \mathbb{R}^+$ such that $\|\mathbf{x}_v - \mathbf{x}_{v'}\|_2 > C$ for all pairs of nodes. Then we have that there exists a parameter setting for Algorithm 1, using a pooling aggregator at depth $k = 1$, where this pooling aggregator has $\geq 2$ hidden layers with rectified non-linear units, such that*

$$\mathbf{h}_v^1 \neq \mathbf{h}_{v'}^1, \forall (v, v') \in \{(v, v') : \exists u \in \mathcal{V}, v, v' \in \mathcal{N}^3(u)\}, \mathbf{h}_v^1, \mathbf{h}_{v'}^1 \in \mathcal{E}_I^{\chi(\mathcal{G}^3)}$$

*where $\mathcal{E}_I^{\chi(\mathcal{G}^3)}$ is the set of one-hot indicator vectors of dimension $\chi(\mathcal{G}^3)$.*

*Proof.* By the definition of the chromatic number, we know that we can label every node in $\mathcal{V}$ using $\chi(\mathcal{G}^3)$ unique colors, such that no two nodes that co-occur in any node's 3-hop neighborhood are assigned the same color. Thus, with exactly $\chi(\mathcal{G}^3)$ dimensions we can assign a unique one-hot indicator vector to every node, where no two nodes that co-occur in any 3-hop neighborhood have the same vector. In other words, each color defines a subset of nodes $\mathcal{D} \subseteq \mathcal{V}$ and this subset of nodes can all be mapped to the same indicator vector without introducing conflicts.

By Lemma 1 and 2 and the assumption that $\|\mathbf{x}_v - \mathbf{x}_{v'}\|_2 > C$ for all pairs of nodes, we can choose an $\epsilon < 0.5$ and there exists a single-layer MLP, $f_{\theta_\sigma}$, such that for any subset of nodes $\mathcal{D} \subseteq \mathcal{V}$:

$$\begin{cases} f_{\theta_\sigma}(\mathbf{x}_v) > 0, & \forall v \in \mathcal{D} \\ f_{\theta_\sigma}(\mathbf{x}_v) < 0, & \forall v \in \mathcal{V} \backslash \mathcal{D}. \end{cases} \tag{9}$$

By making this MLP one layer deeper and specifically using a rectified linear activation function, we can return a positive value only for nodes in the subset $\mathcal{D}$ and zero otherwise, and, since we normalize after applying the aggregator layer, this single positive value can be mapped to an indicator vector. Moreover, we can create $\chi(\mathcal{G}^3)$ such MLPs, where each MLP corresponds to a different color/subset; equivalently each MLP corresponds to a different max-pooling dimension in equation 3 of the main text. □

We now restate Theorem 1 and provide a proof.

**Theorem 1.** *Let $\mathbf{x}_v \in \mathbb{R}^d, \forall v \in \mathcal{V}$ denote the feature inputs for Algorithm 1 on graph $\mathcal{G} = (\mathcal{V}, \mathcal{E})$, where $U$ is any compact subset of $\mathbb{R}^d$. Suppose that there exists a fixed positive constant $C \in \mathbb{R}^+$ such that $\|\mathbf{x}_v - \mathbf{x}_{v'}\|_2 > C$ for all pairs of nodes. Then we have that $\forall \epsilon > 0$ there exists a parameter setting $\Theta^*$ for Algorithm 1 such that after $K = 4$ iterations*

$$|z_v - c_v| < \epsilon, \forall v \in \mathcal{V},$$

*where $z_v \in \mathbb{R}$ are final output values generated by Algorithm 1 and $c_v$ are node clustering coefficients, as defined in [38].*

*Proof.* Without loss of generality, we describe how to compute the clustering coefficient for an arbitrary node $v$. For notational convenience we use $\oplus$ to denote vector concatenation and $d_v$ to denote the degree of node $v$. This proof requires 4 iterations of Algorithm 1, where we use the pooling aggregator at all depths. For clarity and we ignore issues related to vector normalization and we use the fact that the pooling aggregator can approximate any Hausdorff continuous function to an arbitrary $\epsilon$ precision [29]. Note that we can always account for normalization constants (line 7 in Algorithm 1) by having aggregators prepend a unit value to all output representations; the normalization constant can then be recovered at later layers by taking the inverse of this prepended value. Note also that almost certainly exist settings where the symmetric functions described below can be computed exactly by the pooling aggregator (or a variant of it), but the symmetric universal approximation theorem of [29] along with Lipschitz continuity arguments suffice for the purposes of proving identifiability of clustering coefficients (up to an arbitrary precision). In particular, the functions described below, that we need approximate to compute clustering coefficients, are all Lipschitz continuous on their domains (assuming we only run on nodes with positive degrees) so the errors introduced by approximation remain bounded by fixed constants (that can be made arbitrarily small).

We assume that the weight matrices, $\mathbf{W}^1, \mathbf{W}^2$ at depths $k = 2$ and $k = 3$ are the identity, and that all non-linearities are rectified linear units. In addition, for the final iteration (i.e, $k = 4$) we completely ignore neighborhood information and simply treat this layers as an MLP with a single hidden layer. Theorem 1 can be equivalently stated as requiring $K = 3$ iterations of Algorithm 1, with the representations then being fed to a single-layer MLP.

By Lemma 3, we can assume that at depth $k = 1$ all nodes in $v$'s 3-hop neighborhood have unique, one-hot indicator vectors, $\mathbf{h}_v^1 \in \mathcal{E}_I$. Thus, at depth $k = 2$ in Algorithm 1, suppose that we sum the unnormalized representations of the neighboring nodes (which is possible by Lemma 4). Then without loss of generality, we will have that $\mathbf{h}_v^2 = \mathbf{h}_v^1 \oplus \mathbf{A}_v$ where $\mathbf{A}$ is the adjacency matrix of the subgraph containing all nodes connected to $v$ in $G^3$ and $\mathbf{A}_v$ is the row of the adjacency matrix corresponding to $v$. Then, at depth $k = 3$, again assume that we sum the neighboring representations (with the weight matrices as the identity), then we will have that

$$\mathbf{h}_v^3 = \mathbf{x}_v \oplus \mathbf{A}_v \oplus \left( \sum_{v \in \mathcal{N}(v)} \mathbf{x}_v \oplus \mathbf{A}_v \right). \tag{10}$$

Letting $m$ denote the dimensionality of the $\mathbf{h}_v^1$ vectors (i.e., $m \equiv \chi(G^3)$ from Lemma 3) and using square brackets to denote vector indexing, we can observe that

- $\mathbf{a} \equiv \mathbf{h}_v^3[0 : m]$ is $v$'s one-hot indicator vector.

- $\mathbf{b} \equiv \mathbf{h}_v^3[m : 2m]$ is $v$'s row in the adjacency matrix, $\mathbf{A}$.

- $\mathbf{c} \equiv \mathbf{h}_v^3[3m : 4m]$ is the sum of the adjacency rows of $v$'s neighbors.

Thus, we have that $\mathbf{b}^\top \mathbf{c}$ is the number of edges in the subgraph containing only $v$ and it's immediate neighbors and $\sum_{i=0}^m \mathbf{b}[i] = d_v$. Finally we can compute

$$\frac{2(\mathbf{b}^\top \mathbf{c} - d_v)}{(d_v)(d_v - 1)} = \frac{2|\{e_{v,v'} : v, v' \in \mathcal{N}(v), e_{v,v'} \in \mathcal{E}\}|}{(d_v)(d_v - 1)} \tag{11}$$

$$= c_v, \tag{12}$$

and since this is a continuous function of $\mathbf{h}_v^3$, we can approximate it to an arbitrary $\epsilon$ precision with a single-layer MLP (or equivalently, one more iteration of Algorithm 1, ignoring neighborhood information). Again this last step follows directly from [15]. $\qquad\square$

Figure 3: Accuracy (in F1-score) for different approaches on the citation data as the feature matrix is incrementally replaced with random Gaussian noise.

**Corollary 2.** *Suppose we sample nodes features from any probability distribution $\mu$ over $\mathbf{x} \in U$, where $\mu$ is absolutely continuous with respect to the Lebesgue measure. Then the conditions of Theorem 1 are almost surely satisfied with feature inputs $\mathbf{x}_v \sim \mu$.*

Corollary 2 is a direct consequence of Theorem 1 and the fact that, for any probability distribution that is absolutely continuous w.r.t. the Lebesgue measure, the probability of sampling two identical points is zero. Empirically, we found that GraphSAGE-pool was in fact capable of maintaining modest performance by leveraging graph structure, even with completely random feature inputs (see Figure **??**). However, the performance GraphSAGE-GCN was not so robust, which makes intuitive sense given that the Lemmas 1, 2, and 3 rely directly on the universal expressive capability of the pooling aggregator.

Finally, we note that Theorem 1 and Corollary 2 are expressed with respect to a particular given graph and are thus somewhat transductive. For the inductive setting, we can state

**Corollary 3.** *Suppose that for all graphs $\mathcal{G} = (\mathcal{V}, \mathcal{E})$ belonging to some class of graphs $G^*$, we have that $\exists k, d \geq 0, k, d \in \mathbb{Z}$ such that*

$$\mathbf{h}_v^k \neq \mathbf{h}_{v'}^k, \forall (v, v') \in \{(v, v') : \exists u \in \mathcal{V}, v, v' \in \mathcal{N}^3(u)\}, \mathbf{h}_v^k, \mathbf{h}_{v'}^k \in \mathcal{E}_I^d,$$

*then we can approximate clustering coefficients to an arbitrary epsilon after $K = k + 4$ iterations of Algorithm 1.*

Corollary 3 simply states that if after $k$ iterations of Algorithm 1, we can learn to uniquely identify nodes for a class of graphs, then we can also approximate clustering coefficients to an arbitrary precision for this class of graphs.