[Reviews · NeurIPS 2017]

Reviewer 1



The authors introduce GraphSAGE, an inductive learning representation learning method for graph-structured data. Unlike previous transductive methods, GraphSAGE is able to generalize the representation to previously unseen nodes. The representation is learned through a recursive process that samples from a node's neighborhood and aggregates the results. GraphSAGE outperforms other popular embedding techniques at three node classification tasks. Quality: The quality of the paper is very high. The framework has several attractive qualities: a representation that is invariant with respect to node index, strong performance at three inductive node classification tasks, and fast training and inference in practice. The authors include code that they intend to release to the public, which is likely to increase the impact of the work. Clarity: The paper is very well-written, well-organized, and enjoyable to read. Originality: The idea is incremental but creative and useful. Significance: This work is likely to be impactful on the NIPS community due to the strong results as well as the fast and publicly available implementation. The work builds on recent developments to offer a tangible improvement to the community. Overall impression: A high-quality paper that is likely to be of interest to the NIPS community and impactful in the field. Clear accept.

Reviewer 2



The paper proposes an approach for generating node embeddings in large graphs, with the important property that it is applicable to nodes previously not seen (for example, to a node that was created just recently in a dynamic graph). I am not an expert in this area, so I am afraid this review is not particularly well informed. In my view, the problem is important and the approach has many potential applications. The authors provide an extensive review of related work and adequately explain how their approach is different (and the differences are indeed important). Experimental evaluation is rigorous; the results are very positive, clearly demonstrating the utility of the approach. Section 5 is a very nice addition to the paper. It was a welcome relief to see plots where the text is clearly legible (this was in fact the only paper in my batch that did so).

Reviewer 3



The manuscript deals with converting high-dimensional nodes in a graph into low dimensional representations, where it is not necessary to use the whole network (e.g. when nodes are added or removed) to calculate the low dimension representation of a given node. 1. The introduction is not written very clearly, and it takes several "read-through"s to understand the author's goal. For example, they use the term transductive without explaining what it means. It is left to the reader to deduce that they probably mean the setting presented in the first sentence of the previous paragraph. 2. The fact that the manuscript is far from the standard introduction, methods, results framework further detracts from its readability. 3. The relation between the proposed algorithm and the Weisfeiler-Lehman isomorphism test is very interesting and should be explained in more details, and possibly in the background, and not as an afterthought as it is now. It provides both motivation and an explanation as to why there is a chance that the unsupervised embedding may be useful in some context. 4. The extension to the minibatch setting is not clear enough. In a highly connected graph (such as a PPI), the neighborhood set of a small K may already be the whole network, making the task computationally expensive.